# Application of Machine Learning Algorithms to Classification of Pb–Zn Deposit Types Using LA–ICP–MS Data of Sphalerite

**Guo-Tao Sun** [1,2,3,*] **and Jia-Xi Zhou** [4,5]

1   State Key Laboratory of Public Big Data, Guizhou University, Guiyang 550025, China
2   College of Resources and Environmental Engineering, Guizhou University, Guiyang 550025, China
3   Key Laboratory of Karst Georesources and Environment, Ministry of Education, Guiyang 500025, China
4   School of Earth Sciences, Yunnan University, Kunming 650500, China
5   Key Laboratory of Critical Minerals Metallogeny in Universities of Yunnan Province, Kunming 650500, China
*   Correspondence: gtsun@gzu.edu.cn

**Abstract:** Pb–Zn deposits supply a significant proportion of critical metals, such as In, Ga, Ge, and Co. Due to the growing demand for critical metals, it is urgent to clarify the different types of Pb–Zn deposits to improve exploration. The trace element concentrations of sphalerite can be used to classify the types of Pb–Zn deposits. However, it is difficult to assess the multivariable system through simple data analysis directly. Here, we collected more than 2200 analyses with 14 elements (Mn, Fe, Co, Ni, Cu, Ga, Ge, Ag, Cd, In, Sn, Sb, Pb, and Bi) from 65 deposits, including 48 analyses from carbonate replacement (CR), 684 analyses from distal magmatic-hydrothermal (DMH), 197 analyses from epithermal, 456 analyses from Mississippi Valley-type (MVT), 199 analyses from sedimentary exhalative (SEDEX), 377 analyses from skarn, and 322 analyses from volcanogenic massive sulfide (VMS) types of Pb–Zn deposits. The critical metals in different types of deposits are summarized. Machine learning algorithms, namely, decision tree (DT), K-nearest neighbors (KNN), naive Bayes (NB), random forest (RF), and support vector machine (SVM), are applied to process and explore the classification. Learning curves show that the DT and RF classifiers are the most suitable for classification. Testing of the DT and RF classifier yielded accuracies of 91.2% and 95.4%, respectively. In the DT classifier, the feature importances of trace elements suggest that Ni (0.22), Mn (0.17), Cd (0.13), Co (0.11), and Fe (0.09) are significant for classification. Furthermore, the visual DT graph shows that the Mn contents of sphalerite allow the division of the seven classes into three groups: (1) depleted in Mn, including MVT and CR types; (2) enriched in Mn, including epithermal, skarn, SEDEX, and VMS deposits; and (3) DMH deposits, which have variable Mn contents. Data mining also reveals that VMS and skarn deposits have distinct Co and Ni contents and that SEDEX and DMH deposits have different Ni and Ge contents. The optimal DT and RF classifiers are deployed at Streamlit cloud workspace. Researchers can select DT or RF classifier and input trace element data of sphalerite to classify the Pb–Zn deposit type.

**Keywords:** machine learning; sphalerite; LA–ICP–MS; Pb–Zn deposits; web app

## 1. Introduction

Base metal (Pb–Zn) deposits are important sources of critical metals, such as indium (In), germanium (Ge), gallium (Ga), cobalt (Co), and cadmium (Cd). Distinct types of Pb–Zn deposits are enriched in different critical metals due to different sources and ore-forming processes. Distinguishing the Pb–Zn types is essential for identifying new sources of critical metals and enhancing exploration efficiency. Previous studies identified the differences in the trace elements of sphalerite from different types of Pb–Zn deposits [1,2]. However, the studies lack statistical analysis to distinguish the different deposit types reliably. Frenzel et al. [3] applied principal component analysis (PCA) to identify the differences among different types of Pb–Zn deposits. However, PCA is a dimensionality

reduction method and is less effective in classification. Therefore, the study did not show classification for different types of Pb–Zn deposits.

Machine learning (ML) is an effective empirical approach for classifying nonlinear systems. Such systems are massively multivariate, involving a few or literally thousands of variables. The types of ML algorithms for classification mainly include K-nearest neighbor (KNN), decision tree (DT), support vector machine (SVM), artificial neural network (ANN), random forest (RF), and naive Bayes (NB). ML has been widely applied to science and engineering problems, such as data mining, artificial intelligence, DNA sequencing, and pattern recognition. The application of ML in geoscience, especially economic geology, is new and limited [4–7].

Here, we collected over 2200 laser ablation–inductively coupled plasma–mass spectrometry (LA–ICP–MS) analyses of sphalerite from 65 deposits (Figure 1) and applied DT, KNN, NB, RF, and SVM algorithms on the Scikit-learn package in Python to classify the types of Pb–Zn deposits using trace element concentrations of sphalerite. Our contribution is twofold: We provide a statistical classification for different types of Pb–Zn deposits and deploy the classifications app online to be accessed by economic geologists.

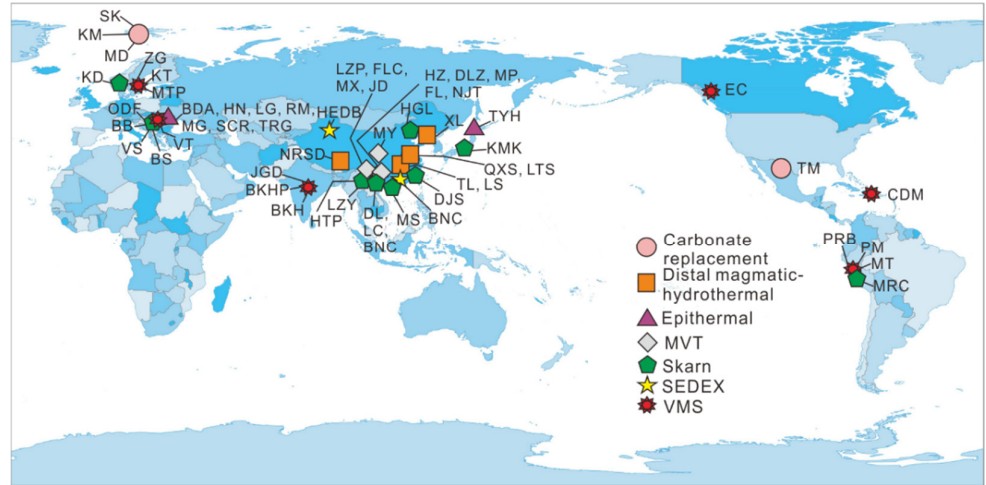

**Figure 1.** Principal Pb–Zn deposits that reported LA–ICP–MS data of sphalerite. CR type: TM, Tres Marias; SK, Sinkholmen; KM, Kapp Mineral; MD, Melandsgruve. DMH type: TL, Taolin; XL, Xinling; LTS, Luotuoshan; NRSD, Narusongduo; QXS, Qixiashan; LS, Lishan. Epithermal type: BDA, Baia de Aries; HN, Hanes; LG, Larga; RM, Rosia Montana; MG, Magura; SCR, Sacaramb; TRG, Toroiaga; TYH, Toyoha. MVT: DLZ, Daliangzi; HZ, Huize; MX, Mengxing; LZP, Liziping; FLC, Fulongchang; NJT, Niujiaotang; JD, Jinding; MP, Maoping; FL, Fule; MY, Mayuan. Skarn type: HTP, Hetaoping; LZY, Luziyuan; ODF, Ocna de Fier; BB, Baita Bihor; VS, Valea Seaca; BS, Baisoara; KD, Konnerudkollen; KMK, Kamioka; DL, Dulong; LC, Laochang; MS, Miaoshan; MRC, Morococha; BNC, Bainiuchang. SEDEX type: DBS, Dabaoshan; HEDB, Haerdaban. VMS type: VT, Vorta; EC, Eskay Creek; ZG, Zinkgruvan; KT, Kaveltorp; MTP, Marketorp; BKHP, Banskhapa; JGD, Jangaldehri; BKH, Biskhan; MT, María Teresa; PRB, Perubar; PM, Palma; CDM, Cerro de Maimón. Note: Xinling both has DMH and epithermal types of mineralization; Morococha has skarn, epithermal and DMH types of mineralization.

## 2. Data Preparation and Packages

Data preparation includes data collection and data preprocessing for statistical analyses. Data collection and preprocessing were primarily conducted in Microsoft Excel.

### 2.1. Data Sources

For data consistency, the collected trace element concentrations of sphalerite were mainly determined by LA–ICP–MS analysis. The LA–ICP–MS data have been collected from published articles [1,2,8–31], leading to a database of 2283 sphalerites from carbonate

replacement (CR), distal magmatic-hydrothermal (DMH), epithermal, Mississippi Valley-type (MVT), sedimentary exhalative (SEDEX), skarn, and volcanogenic massive sulfide (VMS) types of Pb–Zn deposits (Table 1).

**Table 1.** Summary of the collected sphalerite LA–ICP–MS dataset.

| Deposit | Country | Type | Number | References | Deposit | Country | Type | Number | References |
|---|---|---|---|---|---|---|---|---|---|
| Tres Marias | Mexico | CR | 22 | [1] | Mayuan | China | MVT | 50 | [8] |
| Sinkholmen | Norway | CR | 8 | [1] | Hetaoping | China | Skarn | 24 | [7] |
| Kapp Mineral | Norway | CR | 10 | [1] | Luziyuan | China | Skarn | 24 | [7] |
| Melandsgruve | Norway | CR | 8 | [1] | Majdanpek | Serbia | Skarn | 8 | [1] |
| Taolin | China | DMH | 64 | [21] | Ocna de Fier | Romania | Skarn | 37 | [1] |
| Xinling | China | DMH | 25 | [20] | Baita Bihor | Romania | Skarn | 30 | [1] |
| Luotuoshan | China | DMH | 35 | [12] | Valea Seaca | Romania | Skarn | 6 | [1] |
| Narusongduo | China | DMH | 66 | [16] | Baisoara | Romania | Skarn | 20 | [1] |
| Qixiashan | China | DMH | 122 | [9,19] | Lefevre | Canada | Skarn | 8 | [1] |
| Morococha | Peru | DMH | 323 | [28] | Konnerudkollen | Norway | Skarn | 5 | [1] |
| Weilasituo | China | DMH | 22 | [11] | Kamioka | Japan | Skarn | 8 | [1] |
| Lishan | China | DMH | 27 | [21] | Dulong | China | Skarn | 57 | [23] |
| Baia de Aries | Romania | Epithermal | 6 | [1] | Laochang | China | Skarn | 16 | [26] |
| Hanes | Romania | Epithermal | 8 | [1] | Miaoshan | China | Skarn | 10 | [10] |
| Larga | Romania | Epithermal | 8 | [1] | Huanggangliang | China | Skarn | 2 | [13] |
| Rosia Montana | Romania | Epithermal | 20 | [1] | Dingjiashan | China | Skarn | 52 | [27] |
| Magura | Romania | Epithermal | 8 | [1] | Morococha | Peru | Skarn | 52 | [28] |
| Sacaramb | Romania | Epithermal | 11 | [1] | Bainiuchang | China | Skarn | 18 | [7] |
| Toroiaga | Romania | Epithermal | 6 | [1] | Dabaoshan | China | SEDEX | 26 | [7] |
| Toyoha | Japan | Epithermal | 22 | [1] | Haerdaban | China | SEDEX | 173 | [29] |
| Wunuer | China | Epithermal | 82 | [18] | Vorta | Romania | VMS | 8 | [1] |
| Xinling | China | Epithermal | 19 | [20] | Eskay Creek | Canada | VMS | 12 | [1] |
| Morococha | Peru | Epithermal | 7 | [28] | Zinkgruvan | Sweden | VMS | 5 | [1] |
| Daliangzi | China | MVT | 85 | [14] | Kaveltorp | Sweden | VMS | 8 | [1] |
| Huize | China | MVT | 24 | [7] | Marketorp | Sweden | VMS | 8 | [1] |
| Mengxing | China | MVT | 18 | [7] | Sauda Sa | Norway | VMS | 10 | [1] |
| Liziping | China | MVT | 67 | [30] | Banskhapa | Indian | VMS | 5 | [25] |
| Fulongchang | China | MVT | 48 | [30] | Jangaldehri | Indian | VMS | 10 | [25] |
| Angouran | Iran | MVT | 43 | [17] | Biskhan | Indian | VMS | 11 | [25] |
| Niujiaotang | China | MVT | 26 | [7] | María Teresa | Peru | VMS | 141 | [31] |
| Jinding | China | MVT | 24 | [7] | Perubar | Peru | VMS | 50 | [31] |
| Maoping | China | MVT | 49 | [24] | Palma | Peru | VMS | 37 | [31] |
| Fule | China | MVT | 22 | [15] | Cerro de Maimón | Dominican Republic | VMS | 17 | [31] |

### 2.2. Data Preprocessing

Data preprocessing is a process that fills in missing values, such as some analyses lacking several trace element concentrations and some values below the detection limits.

Most samples included the concentrations of Mn, Fe, Co, Ni, Cu, Ga, Ge, Ag, Cd, In, Sn, Sb, Pb, and Bi. The lack of As, Mo, Hg, Se, Bi, and Tl is significant. Therefore, these elements are excluded from the data. Other unanalyzed data were filled based on the mean value for the elements from others of the same type in the dataset. These data were assumed to be reasonable estimates, as these elements are commonly below detection. This method has been used by Gregory et al., (2019) for data preprocessing. When analyses were below the detection limits, either the detection limit was used, or a value based on nearby values was inserted.

### 2.3. Library and Package Preparation

Numpy and Pandas are fundamental Python libraries for scientific computing. Streamlit is an open-access library that can easily create custom web apps for machine learning. We use Scikit-learn, a simple and efficient tool for machine learning in Python for classification. The package splits the raw data and trains and tests the DT and RF classifiers. Graphviz, a graph visualization package, is used to represent the structural information of decision trees. The Streamlit Cloud is a workspace for deploying and managing Streamlit apps.

### 3. Description of ML Methods and Pb–Zn Deposits

*3.1. Description of ML Methods*

In this study, we applied ML methods, including DT, KNN, NB, RF, and SVM, to establish the classifiers. A simple description of these methods is presented below.

The DT is an effective classification method in data mining classification [32]. It is defined as a process that partitions a dataset into smaller classes. The decision rules are based on the tests defined at each branch [33]. The DT comprises three types of nodes: a root node that has no parent node, some internal nodes (splits) that have both parent and descendant nodes, and a set of terminal nodes (leaves) with no descendant nodes [34].

The KNN method is based on the spatial similarity between a test sample and its $k$ neighbors [35]. The distance is computed in the feature space from the test sample to each sample for which the label is known. The estimate of the test sample is based on the label of $k$-nearest samples [36]. Therefore, the parameter $k$ is the most important for KNN. As a local method, the KNN is known to be strong in the case of large data and low dimensions [37].

NB is a special form of Bayesian network that is one of the most effective theoretical models in the field of uncertain knowledge expression and reasoning. NB assumes that all variables are mutually independent [38]. Given an unclassified sample $x$ with features $(a_1, a_2, \ldots, a_m)$ and a labeled class $C$ with members $(y_1, y_2, \ldots, y_n)$, if $P(y_k \,|\, x) = \max \{P(y_1 \,|\, x), P(y_2 \,|\, x), \ldots, P(y_n \,|\, x)\}$, $x$ belongs to $y_k$. According to the Bayesian principle, the following derivation is made: $P(y_k \,|\, x) = P(x \,|\, y_k)\, P(y_k)/P(x)$. NB can further simplify the calculation process to $P(x \,|\, y_k)*P(y_k) = P(a_1 \,|\, y_k)\, P(a_2 \,|\, y_k), \ldots, P(a_m \,|\, y_k)\, P(y_k)$ [38].

RF is an ensemble ML algorithm that combines a set of decision trees for classification and prediction [39]. A number of features are randomly chosen for a single DT. The bootstrapping method randomly chooses training data for a single DT. The examples are classified by taking a majority vote cast from all the DT predictors [40].

The SVM method is based on statistical learning theory and is used to determine the location of decision boundaries (optimal hyperplane) that maximizes the distance between the classes [41]. The support vector machine can be linear or nonlinear. In a binary classification problem where classes are linearly separable, the hyperplane corresponds to a linear boundary, and the SVM selects the boundary that produces the maximum margin between the two classes [42]. If the binary classification problem is not linearly separable, the SVM is designed to identify a hyperplane (e.g., plane and sphere) that maximizes the margin. For nonlinear SVM problems, kernel functions, such as polynomial and sigmoid functions, are used to reduce the computational cost of dealing with high-dimensional space by adding an additional dimension to the data [40,41].

*3.2. Description of Deposits and Samples*

The LA–ICP–MS data were collected from 65 deposits worldwide (Table 1). The origin types and references of each deposit are shown in Supplementary Table S1. The dataset comprises 48 analyses from four CD deposits, 684 analyses from eight DMH deposits, 197 analyses from eleven epithermal deposits, 456 analyses from eleven MVT deposits, 377 analyses from seventeen skarn deposits, 199 analyses from two SEDEX deposits, and 322 analyses from thirteen VMS deposits.

### 4. Results

*4.1. Learning Curves*

The learning curves of DT, KNN, NB, RF, and SVM are shown in Figure 2. The learning curve of the DT classifier shows that the accuracy scores of the training data are high, whereas the accuracy scores of cross-validation increase with the increase in the amount of training data (Figure 2a,b). Because the training scores are higher than the cross-validation scores, the classifier is overfitting. The learning curves of KNN increase with increasing training data size. When the training data size is above 1500, the cross-validation scores are up to 0.8 (Figure 2c). The learning curves of the NB classifier converge to 0.38 (Figure 2d), indicating that this classifier is underfitting. The learning curves of the

RF classifier show similar characteristics to those of the DT classifier. The performance of cross-validation is better than that of DT. The cross-validation scores reach 0.95 when the scale of training data is up to 1500 (Figure 2e,f). The learning curves of SVM show that the classifier performs better when the training data are above 1200. The cross-validation scores of this classifier are nearly 0.65 (Figure 2g). The learning curves show that the RF classifier has the best performance for the dataset, whereas the NB has the worst performance.

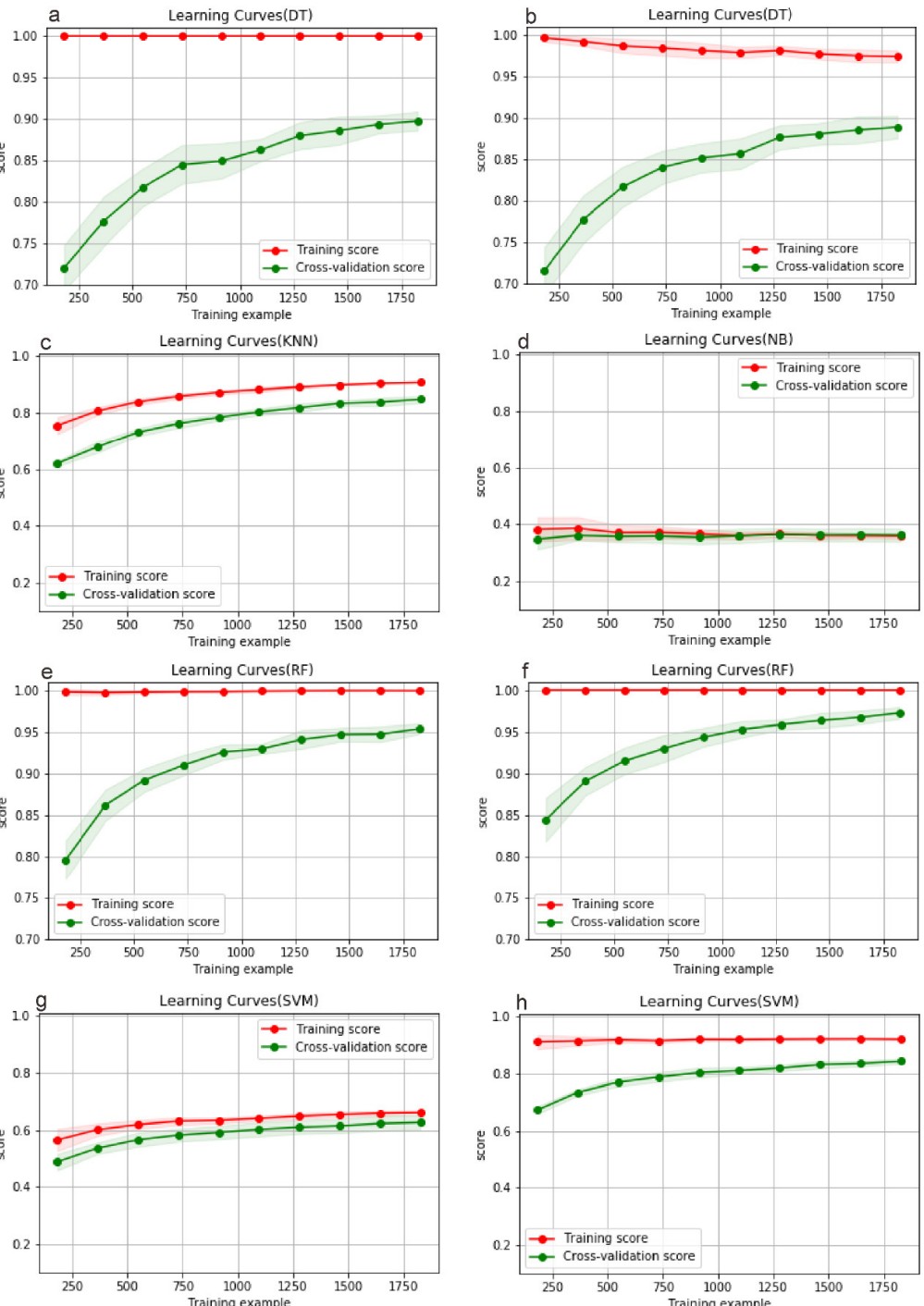

**Figure 2.** The learning curves of ML-classifiers. (**a**) decision tree classifier, (**b**) decision tree classifier after tuning hyperparameters, (**c**) K-nearest neighbors classifier, (**d**) naive Bayes classifier, (**e**) random forest classifiers, (**f**) random forest classifier after tuning hyperparameters, (**g**) support vector machine classifier, (**h**) support vector machine classifier after tuning hyperparameters.

The DT and RF classifiers have different degrees of overfitting, whereas the KNN, NB, and SVM classifiers have different degrees of underfitting. The scores of the underfitting classifiers cannot be improved by increasing the size of the training dataset. The tuning hyperparameters can somewhat improve the accuracies of the underfitting classifiers. For example, the grid search techniques found that the optimal SVM classifier had an accuracy of up to 0.85 (Figure 2h). The overfitting classifiers can be optimized by tuning the hyperparameters and increasing the size of the dataset. Therefore, the DT and RF classifiers are further tuning the hyperparameters by grid search techniques. Parameter optimization decreases the overfitting or improves the overall accuracies of the classifiers (Figure 2b,f).

*4.2. Feature Importances*

Feature importances are defined as the total decrease in node impurity. If the value is low, then the feature is not important, and vice versa. The feature importances of elements used for ML methods are shown in Figure 3. Among them, the feature importances of Ni (0.22), Mn (0.17), Cd (0.13), Co (0.11), and Fe (0.09) are higher than those of other elements, suggesting that Ni, Mn, Cd, Co, and Fe are effective in classifying the types of deposits.

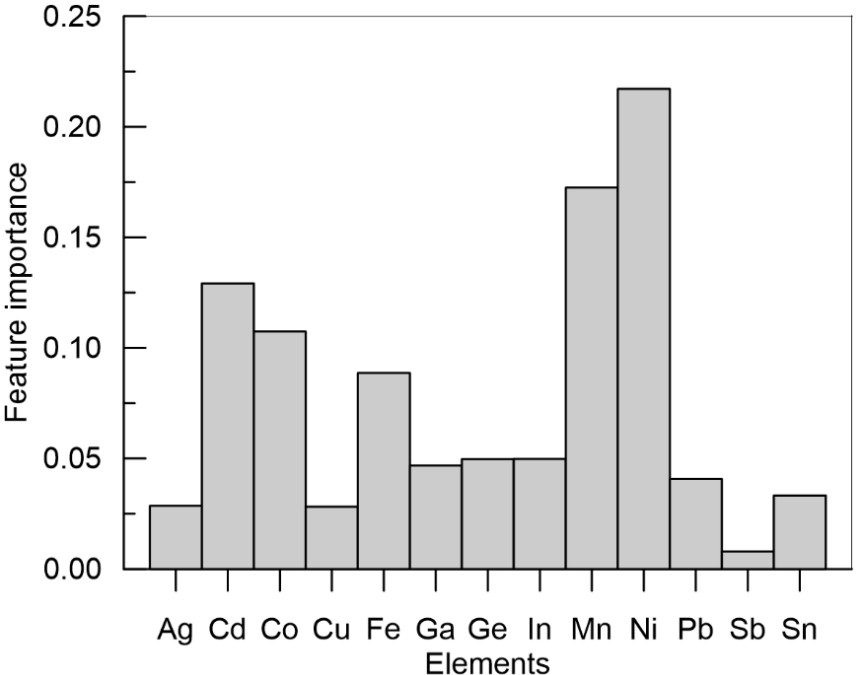

**Figure 3.** Feature importances of each element that was used in classifiers.

*4.3. Accuracies of the DT and RF Classifiers*

As presented above, the DT and RF algorithms show good performance for the classification problem. Therefore, the two algorithms are further assessed based on other parameters, such as accuracy score, precision, and recall. The accuracy scores of the DT and RF classifiers are listed in Table 2. In this work, 70% of the data are used as training data to produce the models, whereas the remaining data are used to test the performance of the models. These classifiers were run 10 times with random selections of training data and testing data to assess the effectiveness of the classifiers. The accuracy scores of the DT classifier are between 0.871 and 0.907, with a mean of 0.890 and a standard deviation (SD) of 0.011. The accuracy scores of the RF classifier range from 0.953 to 0.976, with a mean of 0.969 and an SD of 0.007. The accuracy scores show that the RF classifier is effective in distinguishing the different types of deposits.

**Table 2.** Precision, recall, and F1 score of classifiers.

| Classifiers | | DT Classifier | | | | | | | RF Classifier | | | | | | | |
|---|---|---|---|---|---|---|---|---|---|---|---|---|---|---|---|---|
| Types | | CR | DMH | Epithermal | MVT | SEDEX | Skarn | VMS | | CR | DMH | Epithermal | MVT | SEDEX | Skarn | VMS |
| Percision | 1 | 0.353 | 0.835 | 0.830 | 0.872 | 0.917 | 0.795 | 0.969 | 1 | 1.000 | 0.976 | 0.966 | 0.935 | 1.000 | 0.975 | 1.000 |
| | 2 | 0.692 | 0.921 | 0.825 | 0.917 | 0.963 | 0.870 | 0.960 | 2 | 1.000 | 0.975 | 0.966 | 0.963 | 1.000 | 0.968 | 0.990 |
| | 3 | 0.571 | 0.932 | 0.764 | 0.919 | 0.923 | 0.826 | 0.854 | 3 | 1.000 | 0.986 | 0.965 | 0.956 | 1.000 | 0.928 | 0.989 |
| | 4 | 0.750 | 0.911 | 0.800 | 0.886 | 0.849 | 0.860 | 0.906 | 4 | 1.000 | 0.952 | 1.000 | 0.914 | 0.980 | 0.972 | 0.989 |
| | 5 | 0.667 | 0.868 | 0.818 | 0.954 | 0.963 | 0.898 | 0.920 | 5 | 1.000 | 0.941 | 0.948 | 0.970 | 1.000 | 0.939 | 0.989 |
| | Mean | 0.607 | 0.894 | 0.807 | 0.910 | 0.923 | 0.850 | 0.922 | Mean | 1.000 | 0.966 | 0.969 | 0.948 | 0.996 | 0.956 | 0.992 |
| | SD | 0.139 | 0.037 | 0.024 | 0.028 | 0.042 | 0.036 | 0.041 | SD | 0.000 | 0.017 | 0.017 | 0.021 | 0.008 | 0.019 | 0.004 |
| Recall | 1 | 0.375 | 0.921 | 0.650 | 0.848 | 0.902 | 0.789 | 0.939 | 1 | 0.563 | 0.990 | 0.950 | 1.000 | 1.000 | 0.953 | 0.970 |
| | 2 | 0.600 | 0.907 | 0.825 | 0.905 | 0.963 | 0.934 | 0.941 | 2 | 0.667 | 0.990 | 0.889 | 1.000 | 1.000 | 0.984 | 0.980 |
| | 3 | 0.571 | 0.894 | 0.689 | 0.895 | 0.923 | 0.905 | 0.936 | 3 | 0.571 | 0.986 | 0.902 | 1.000 | 0.969 | 0.981 | 0.979 |
| | 4 | 0.429 | 0.939 | 0.727 | 0.918 | 0.918 | 0.860 | 0.897 | 4 | 0.381 | 1.000 | 0.818 | 0.994 | 0.980 | 0.991 | 0.969 |
| | 5 | 0.750 | 0.952 | 0.652 | 0.901 | 1.000 | 0.882 | 0.939 | 5 | 0.625 | 0.990 | 0.797 | 0.988 | 1.000 | 0.973 | 0.949 |
| | Mean | 0.545 | 0.923 | 0.709 | 0.893 | 0.941 | 0.874 | 0.930 | Mean | 0.561 | 0.991 | 0.871 | 0.996 | 0.990 | 0.976 | 0.969 |
| | SD | 0.133 | 0.021 | 0.065 | 0.024 | 0.036 | 0.049 | 0.017 | SD | 0.098 | 0.005 | 0.056 | 0.005 | 0.013 | 0.013 | 0.011 |
| F1-score | 1 | 0.364 | 0.876 | 0.729 | 0.860 | 0.909 | 0.792 | 0.954 | 1 | 0.720 | 0.983 | 0.958 | 0.967 | 1.000 | 0.967 | 0.985 |
| | 2 | 0.643 | 0.914 | 0.825 | 0.911 | 0.963 | 0.901 | 0.950 | 2 | 0.800 | 0.982 | 0.926 | 0.981 | 1.000 | 0.976 | 0.985 |
| | 3 | 0.571 | 0.913 | 0.724 | 0.907 | 0.923 | 0.864 | 0.893 | 3 | 0.727 | 0.986 | 0.932 | 0.977 | 0.984 | 0.954 | 0.984 |
| | 4 | 0.545 | 0.925 | 0.762 | 0.902 | 0.882 | 0.860 | 0.902 | 4 | 0.552 | 0.975 | 0.900 | 0.952 | 0.980 | 0.981 | 0.979 |
| | 5 | 0.706 | 0.908 | 0.726 | 0.927 | 0.981 | 0.890 | 0.929 | 5 | 0.769 | 0.965 | 0.866 | 0.979 | 1.000 | 0.955 | 0.969 |
| | Mean | 0.566 | 0.907 | 0.753 | 0.901 | 0.932 | 0.861 | 0.926 | Mean | 0.714 | 0.978 | 0.916 | 0.971 | 0.993 | 0.967 | 0.980 |
| | SD | 0.116 | 0.017 | 0.039 | 0.022 | 0.036 | 0.038 | 0.025 | SD | 0.086 | 0.008 | 0.031 | 0.011 | 0.009 | 0.011 | 0.006 |

The confusion matrix shows the prediction results and actual types of deposits (Figure 4). Precision, recall, and F1 scores can be calculated from the confusion matrix, and they can evaluate the classification for individual ore deposit types. Precision is defined as the ratio of correctly predicted samples to predicted samples. The recall is defined as the ratio of correctly predicted samples to actual samples. The F1 score is a measurement of precision and recall. The calculation of the F1 score is shown in Formula (1). The precision, recall, and F1 score are calculated five times with random selections of test data. The precision values for individual types of the DT range from $0.607 \pm 0.139$ to $0.923 \pm 0.042$. Carbonate replacement, DMH, MVT, SEDEX, skarn, and VMS test data were predicted with precisions of $0.607 \pm 0.139$, $0.894 \pm 0.037$, $0.807 \pm 0.024$, $0.910 \pm 0.028$, $0.923 \pm 0.042$, $0.850 \pm 0.036$, and $0.922 \pm 0.041$ on average, respectively. The recall mean values of the predicted DMH, epithermal, MVT, SEDEX, skarn, and VMS test data are $0.545 \pm 0.133$, $0.923 \pm 0.021$, $0.709 \pm 0.065$, $0.893 \pm 0.024$, $0.941 \pm 0.036$, $0.874 \pm 0.049$, and $0.930 \pm 0.017$, respectively. The corresponding F1 scores are $0.566 \pm 0.116$, $0.907 \pm 0.017$, $0.753 \pm 0.039$, $0.901 \pm 0.022$, $0.932 \pm 0.036$, $0.861 \pm 0.038$, and $0.926 \pm 0.025$ on average, respectively. The results show that the DT classifier has low precision, recall, and F1 score for carbonate replacement deposits.

$$\text{F1 score} = 2 \times \text{Precision} \times \text{Recall}/(\text{Precision} + \text{Recall}) \tag{1}$$

The precision for individual types of the RF ranges from $0.948 \pm 0.021$ to $0.92 \pm 0.04$. Carbonate replacement, DMH, epithermal, MVT, SEDEX, skarn, and VMS test data were predicted with precisions of $1.000$, $0.966 \pm 0.017$, $0.969 \pm 0.017$, $0.948 \pm 0.021$, $0.996 \pm 0.008$, $0.956 \pm 0.019$, and $0.992 \pm 0.004$ on average, respectively. The mean values of recall are $0.561 \pm 0.098$, $0.991 \pm 0.005$, $0.871 \pm 0.056$, $0.996 \pm 0.005$, $0.990 \pm 0.013$, $0.976 \pm 0.013$, and $0.969 \pm 0.011$, respectively. The F1 mean scores are $0.714 \pm 0.086$, $0.978 \pm 0.008$, $0.916 \pm 0.031$, $0.971 \pm 0.011$, $0.993 \pm 0.009$, $0.967 \pm 0.011$, and $0.980 \pm 0.006$, respectively. The results show that the RF classifier has higher prediction accuracies for individual types than the DT classifier.

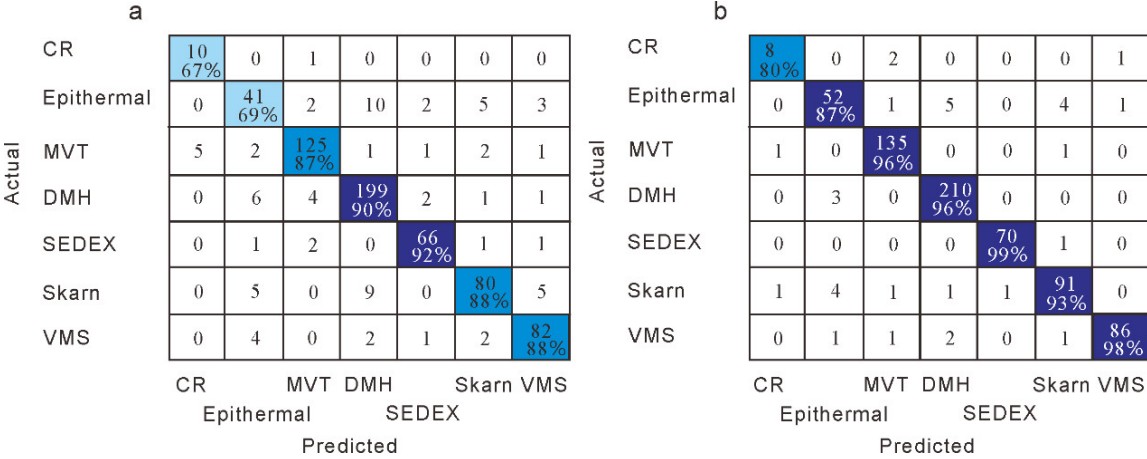

**Figure 4.** Confusion matrix for decision tree (**a**) and random forest (**b**) classifiers of test data. The numbers in the squares are the numbers of analyses and the percentages are precision values.

## 5. Discussion

### 5.1. Critical Metals in Sphalerite

It is well documented that sphalerite is a significant host mineral for critical metals such as Ga, Ge, Cd, In, Co, and Sn [1–3,43–47]. Here, the LA–ICP–MS dataset is firstly used to summarize the critical metal concentrations of sphalerite from different deposit types to investigate the special enrichment of critical metals.

Gallium is a by-product of some MVT deposits. Previous studies have indicated that some MVT deposits produce some Ga metal [46,48]. In this study, sphalerite from some CR,

DMH, epithermal, and VMS deposits also shows enrichment in Ga (Figure 5a), with mean values of 23.1 ppm, 70.5 ppm, 130 ppm, and 27.6 ppm, respectively. Among these types of deposits, high Ga contents are mainly reported from epithermal deposits (Rosia Montana, Romania, 1137 ppm; Sacaramb, Romania, 1126 ppm; Toyoha, Japan, 601 ppm; Xinling, China, 426 ppm; and Morococha, Peru, 1739 ppm) and DMH (Taolin, China, 649 ppm; Morococha, Peru, 2118 ppm; and Lishan, China, 381 ppm) (Supplementary Table S1).

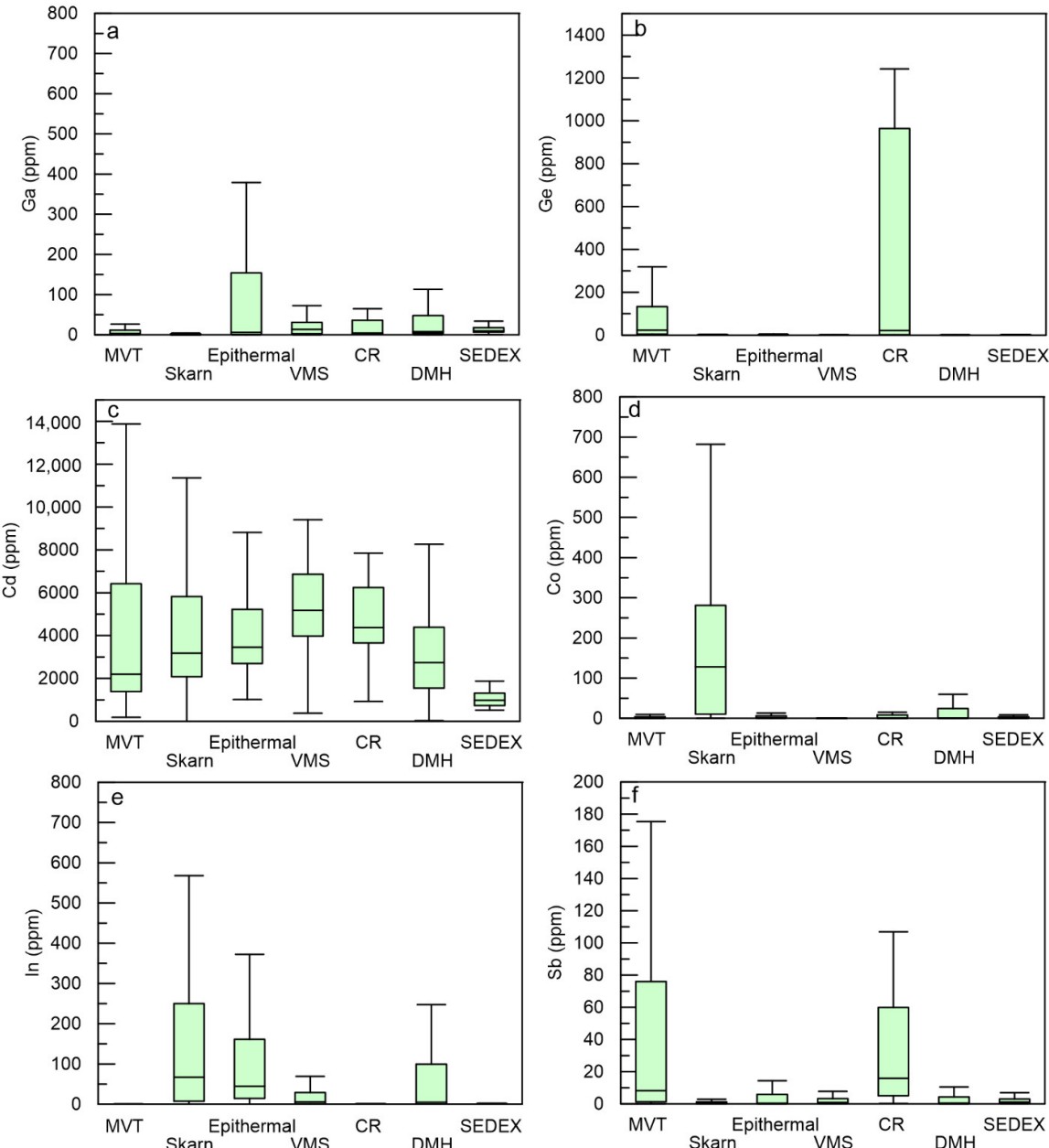

**Figure 5.** Critical metals in sphalerite from different types of deposits.

Germanium is considered to be mainly hosted in MVT sphalerite (Figure 5b). The Mayuan MVT deposit in China has the highest reported Ge contents (1305 ppm), with a mean value of 609 ppm [8]. Other MVT deposits, such as Dalingzi (China), Huize (China), Angouran (Iran), Niujiaotang (China), Maoping (China), and Fule (China), also have high Ge contents of 328, 354, 339, 288, 652, and 941 ppm, respectively (Supplementary Table S1). Notably, the Tres Marias CR deposits in Mexico show Ge contents ranging from 174 to 1242 ppm with a mean value of 704 ppm [1].

Cadmium is enriched in each type of Pb–Zn deposit (Figure 5c). The mean Cd values of sphalerite from the MVT, skarn, epithermal, VMS, CR, DMH, and SEDEX deposits are 5059 ppm, 5960 ppm, 4436 ppm, 5191 ppm, 4361 ppm, 3115 ppm, and 1572 ppm, respectively. The highest Cd content is reported from the Niujiaotang MVT deposit in China (23400 ppm) [7].

Cook et al. [1] reported that cobalt prefers to be enriched in sphalerite from skarn deposits. In the dataset, the Co contents of sphalerite from skarn deposits have a mean value of 237 ppm, which is higher than those of other types (Figure 5d). The highest Co contents of sphalerite are reported from the Ocna de Fier, Romania (2828 ppm) and Konnerudkollen, Norway (1585 ppm) deposits [1]. Some DMH deposits also reported some sphalerite analyses with high Co contents, such as Taolin, China (830 ppm) and Narusongduo, China (370 ppm) (Supplementary Table S1).

Indium has been reported to be mainly enriched in skarn and VMS deposits [1,48–53]. In the dataset, skarn, epithermal, VMS, and DMH deposits have higher In contents than other types (Figure 5e), with mean values of 225, 2732, 31.7, and 120 ppm. The highest In contents (64818 ppm) are reported from the Toyoha epithermal deposit in Japan [1], resulting in an extremely high mean value. The skarn deposits, such as Baita Bihor, Romania (867 ppm), Dulong, China (4572 ppm), Laochang, China (723 ppm), and Miaoshan, China (562 ppm), report In contents higher than 500 ppm [1,10,23,26]. The DMH deposits, including the Qixiashan, China (794 ppm) and Morococha, Peru (1804 ppm) deposits, have In contents higher than 500 ppm [19,28].

Antimony is mainly enriched in sphalerite from MVT and CR deposits (Figure 5f) with mean values of 103 and 43.3 ppm. The highest Sb contents are reported from the Eskay Creek VMS deposit in Canada (117467 ppm) [1]. For MVT deposits, the contents in the Furongchang, China (max 1131 ppm) [30] and Fule, China (max 1403 ppm) [15] deposits are higher than those of other deposits.

Overall, the critical metals in the MVT, skarn, epithermal, VMS, CR, DMH, and SEDEX deposits are Ge–Cd–Sb, Cd–Co–In, Ga–Cd–In, Ga–Cd–In, Ga–Ge–Cd–Sb, Ga–Cd–In, and Cd, respectively (Figure 5). Due to a lack of data, the dataset does not include data from some important deposits, such as Red Dog, leading to the summary being incomplete. Although the dataset has shortcomings, the suggestion that critical metals can correspond with further ML classifiers could somewhat facilitate exploration for critical metals.

*5.2. Assessment of Different ML Methods for Sphalerite LA–ICP–MS Data*

As shown in the learning curves, the NB classifier is the worst for distinguishing the different types of deposits (Figure 2d). The mechanism of NB may result in poor performance. The NB method is based on the hypothesis that the features are unrelated. However, some trace elements in sphalerite are related. For example, Fe and Mn are reported to be negatively related [54]. Correlations between Cu and Ge, Cu and In, and Ag and Sn are common in some deposits due to coupled substitution [1,2,23,55–57]. Therefore, the NB method may not be suitable for the sphalerite trace element data.

The SVM classifier has accuracies of approximately 0.6, as shown in Figure 2g. The problem may be due to the shortcomings of the SVM method. The classical SVM algorithm was originally designed for binary classification [42]. The multiclass classification in this study needs to be solved by combining several binary SVM classifiers, such as one-against-one and one-against-rest [58]. The one-against-rest method was applied in this study. The hyperparameters *C* and *gamma* are significant for the accuracy of the SVM classifier. The hyperparameter *C* is a penalty coefficient for misclassified samples. Higher *C* values will lead to fewer misclassified samples, narrower margins, and higher accuracies. The hyperparameter *gamma* represents the influence distance of the samples. Higher *gamma* values will result in smaller influence distances and higher accuracies. Therefore, we increase the values of *C* and *gamma* values. The learning curves of the optimal SVM (*C* = 10, *gamma* = 0.5) show that the accuracies are up to 0.85 and are better than those of the original SVM classifier (*C* = 1, *gamma* = auto).

The KNN algorithm has a poor effect in the case of unbalanced data, which easily results in misclassification. The numbers of analyses from carbonate replacement, DMH, epithermal, MVT, SEDEX, skarn, and VMS deposits are 48, 684, 197, 527, 199, 377, and 322, respectively, suggesting that the classes in this study are unbalanced. This may explain the accuracies being lower than 0.90 (Figure 2c). The accuracies may be improved by adding the analyses from carbonate replacement, epithermal, and SEDEX deposits.

The DT and RF classifiers produce better accuracies than other classifiers. The investigation for individual types finds that the predictions of the DT classifier for carbonate replacement deposits reveal that the precision and recall are lower than 0.65. The predictions of the RF classifier for carbonate replacement deposits reveal that the recall is lower than 0.60. The small size of the carbonate replacement data may lead to worse precisions and recalls. Further work can increase the size of the carbonate replacement data to improve accuracy.

According to the learning curves, the NB algorithm has the lowest accuracy (<0.4) and is not suitable for the classification of deposits based on trace element data. The SVM, KNN, and DT classifiers have accuracies between 0.8 and 0.9. They can be improved by modifying the data structure. The RF algorithm has the highest accuracies (>0.95) and may be the most suitable for the case in this study.

### 5.3. Statistical Element Characteristics of Different Types of Pb–Zn Deposits

Previous researchers have noticed that trace element concentrations of sphalerite are variable in different types of Pb–Zn deposits. For example, sphalerite from MVT deposits is enriched in Ga, Ge, and Cd, whereas sphalerite from skarn deposits is enriched in In. However, statistical analyses are rarely conducted on the trace elements of sphalerite. Here, the DT graph shows the statistical characteristics of trace elements (Mn, Fe, Co, Ni, Cu, Ga, Ge, Ag, Cd, In, Sn, Sb, Pb, and Bi) for classification (Supplementary Figure S1). For example, Mn is significant for distinguishing the different types of deposits. A total of 92% of analyses from MVT deposits and 94% of analyses from carbonate replacement deposits have Mn contents lower than 62.8 ppm, whereas 96% of skarn, 91% of SEDEX, 100% of VMS, and 99% of epithermal deposits have Mn contents higher than 62.8 ppm. Seventy percent of DMH deposits have values higher than 62.8 ppm. Several explanations can be invoked for the difference in Mn contents, such as formation temperatures and metal sources. Although the MVT and SEDEX deposits may form under similar temperatures [59], the SEDEX deposits have Mn contents higher than 62.8 ppm. Temperature is unlikely to be the reason for the difference. The epithermal, skarn, and VMS deposits mainly form from magmatic-hydrothermal fluids. The MVT deposits are considered to be unrelated to magmatic-hydrothermal fluids [60,61]. Therefore, metal sources may cause different Mn contents. The high Mn contents of sphalerite from SEDEX deposits may also result from different sources.

For the DMH deposits, the Mn contents of sphalerite are variable at the deposit, generation, and sample scales. At the deposit scale, sphalerite with low Mn contents (< 62.8 ppm) forms as the second or third generation in the deposit. For example, the first generation of sphalerite from the Taolin deposit in China has Mn contents between 63.8 and 549 ppm [21], whereas the second generation of sphalerite has Mn contents mainly between 12.9 and 58.4 ppm (Supplementary Table S1). The variance may result from fluid evolution. On a generation scale, the different samples from the second generation show variable Mn contents. For example, some samples of second-generation sphalerite from the Morococha district in Peru show high Mn contents (520–1949 ppm), whereas some samples of the second sphalerite show low Mn contents (<1.4–22 ppm) [28]. Sphalerite can be zoned, showing variable Mn concentrations in the same crystal. On a sample scale, the Mn contents of the second sphalerite can range from <1.4 to 1830 ppm [28]. The Mn contents of sphalerite at the deposit scale may reflect that the Mn concentrations of first-stage ore-forming fluids are high, whereas the Mn concentrations can be low due to the evolution of fluids or mixing with meteoric water. The variance in the generation scale may

be caused by heterogeneous fluids. The heterogeneous Mn contents at the sample scale may result from self-organization processes [54,62].

The data mining reveals that sphalerite from the skarn and VMS deposits has distinct Co and Ni contents. Compared with that from skarn deposits, sphalerite from the VMS deposits has low Co (<33 ppm) and high Ni contents (>0.85 ppm). Although they both occur in magmatic-hydrothermal systems, the metal sources that control the composition of the orebodies may be different. The metal sources of most VMS deposits are from two sources: i) high-temperature reaction zones and ii) magmatic fluids [63]. The high-temperature reaction zones can release abundant Ni from ferromagnesian minerals [64]. The input of magmatic fluids may lead to partial enrichment in Co [65]. The metals of skarn deposits are mainly from felsic magmatic fluids [66,67], which result in significant enrichment in Co and depletion in Ni.

Data mining also finds that the sphalerite from SEDEX has higher Ni (>0.16 ppm) and Ge (>0.86 ppm) contents than that from DMH deposits. It is well documented that pyrite from SEDEX deposits has high Ni contents [5,42,68], and sphalerite from some SEDEX deposits (Red Dog) has Ge contents of approximately 100 ppm [48]. As discussed above, the data-driven ML method can discover the intrinsic structures of data, which are proven to be reasonable by geochemical features. Therefore, ML methods are suitable for exploring the statistical characteristics of geochemical data.

*5.4. Sphalerite Prediction Application*

Utilizing the Streamlit cloud workspace, we deploy the prediction app at https://share.streamlit.io/sun199908/sphalerite--prediction/main/sp-pr-app.py (assecced on 1 September 2022). Two ML algorithms (DT and RF) are provided for testing. Users can select the algorithms and input sphalerite trace element data in the sidebar. The "predict" button is used to start the prediction and display the results. The web app can be used to suggest the origin of some deposits that are debated or newly discovered by drilling. Because different origins of deposits have distinct mineralization regularity, a quick judgment of the origin of deposits is essential to guide further exploration. For example, the origin of Laochang Pb–Zn–Ag–Cu deposit in SW China is debated between VMS [69,70] and magmatic-hydrothermal mineralization [71,72]. The trace element data of sphalerite from the Laochang deposit [70] are inputted into the app. The web app automatically predicts that the deposit may be skarn in origin, which is consistent with the geochronologic evidence (the Re–Os age of pyrite and U–Pb age of hydrothermal titanite are consistent with the zircon U–Pb age) [71]. Then the exploration industry can explore the deposit as a skarn type rather than a VMS type. Furthermore, classification of origin can also timely indicate which critical metals in the Pb–Zn deposits may be recovered by the exploration industry.

Although the application can provide online services, the current version has two shortcomings. First, some important deposits were not included in the dataset, such as the Red Dog deposit in the USA or Mt. Isa in Australia, due to few reports or lack of access. The lack of some data may lead to a partially subjective classification model. The application cannot perform well for all Pb–Zn deposits worldwide. Second, the application only considers the trace elements of sphalerite, which is one aspect of ore genesis. Ore genesis can also be reflected by other geochemical data, such as formation temperature, salinity, and sulfur isotopic composition. These data will be involved in the prediction application in future versions. Furthermore, the host rocks, structure, and other geological characteristics can be transformed into available data and included in future models, as these characteristics could be useful for distinguishing ore deposit types.

## 6. Conclusions

Based on the trace element (Mn, Fe, Co, Ni, Cu, Ga, Ge, Ag, Cd, In, Sn, Sb, and Pb) contents of sphalerite, the DT, KNN, NB, RF, and SVM algorithms were applied to train classifiers that distinguish ore deposit type. The RF algorithm is most suitable for the classification case, with an overall accuracy of $0.969 \pm 0.007$. The significant critical metals

hosted in different types of deposits are summarized based on our dataset. The data mining reveals three statistical characteristics of the trace element data of sphalerite: (1) carbonate replacement and MVT deposits mainly have Mn contents lower than 62.8 ppm, whereas epithermal, SEDEX, skarn, and VMS deposits have Mn contents higher than 62.8 ppm; (2) compared to skarn deposits, VMS deposits have lower Co and higher Ni contents; and (3) compared to DMH deposits, SEDEX deposits have higher Ni and Ge contents. To enable economic geologists to access predictions online, a web app has been created and deployed at https://share.streamlit.io/sun199908/sphalerite-prediction/main/sp-pr-app.py, accessed on 1 September 2022.

**Supplementary Materials:** The following supporting information can be downloaded at: https://www.mdpi.com/article/10.3390/min12101293/s1. Figure S1: Decision tree graph showing the decision processes. Gini values and the color of nodes represent the degree of confusion. The smaller Gini values and deeper color mean a lower degree of confusion. Values present the numbers of analyses from CR, epithermal, MVT, DMH, SEDEX, skarn, VMS at the nodes. Table S1: Complete LA–ICP–MS data of sphalerite.

**Author Contributions:** Investigation, G.-T.S.; Methodology, G.-T.S.; Writing—original draft, G.-T.S.; Writing—review & editing, J.-X.Z. All authors have read and agreed to the published version of the manuscript.

**Funding:** This study was funded by the National Natural Science Foundation of China (42263010, 42202086), the Applied Basic Research Foundation of Yunnan Province (202001BB050020), and the Natural Science Special (special post) scientific research fund project of Guizhou University (No. 2022-24).

**Acknowledgments:** We would like to thank Kai Luo and Hao Zhang, Min Wang, Ni Peng, Ruifeng Zhu, Ye He, Yunlin An, Zhimou Yang at Yunnan University for their assistance in collecting trace element data of sphalerite. We are grateful to Zhilong Huang and Lin Ye for their suggestions.

**Conflicts of Interest:** The authors declare that they have no conflicts of interest or competing interests in the publication of this work.

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
