# Peer review of "Application of Machine Learning Algorithms to Classification of Pb–Zn Deposit Types Using LA–ICP–MS Data of Sphalerite"

_minerals, doi:10.3390/min12101293_

Round 1
Reviewer 1 Report (Previous Reviewer 1)
Dear authors made substantial modifications to the manuscript according to the suggestions. They also provided integration where needed fully replying to my comments and questions.
Author Response
Thanks a lot.
Reviewer 2 Report (Previous Reviewer 2)
Dear Authors,
I have skimmed the paper, and I see that you have significantly improved your presentation with Abstract and Figure 1. This is a useful contribution to researchers and geologists who are interested in area of Pb-Zn deposits.
Author Response
Thanks a lot.
This manuscript is a resubmission of an earlier submission. The following is a list of the peer review reports and author responses from that submission.
Round 1
Reviewer 1 Report
Dear Authors,
This work is an interesting contribution. The work is well written and clear, although it could be improved at some points by adding explanations where missing.
In detail, I would suggest
a) to add a more accurate description of the ML methods used, hence providing a better comprehension to the readers, who may be interested in the topic but are not familiar with it.
b) to further explain why some ML methods work and others do not.
c) to add the localities of the studied deposits.
d) to add a table with the reference study and locality
e) to specify better which elements were used for classifications if all or only the features importances elements
f) to add some information on figures and captions
Further comments are in the pdf text.
The main concerns are about
a) Paragraph 4.1. : It must be integrated with missing information; the learning curves' process should be explained in more detail. Also, it should be explained which hyperparameters were used and why. At the moment, some explanation is in the discussion but is confusing, and some information is missing (more details are in the pdf file).
b)Paragraph 2.2: Preprocessing step: When the values of an element in literature miss, the authors use an average value of elements in a similar deposit type to fill the missing data. This would lead to a simulated dataset that the authors use for their ML calculations.
The preprocessing step must be better explained, and the authors should describe and report the reliability of the classification method.
Further comments are in pdf text
